# Hallucination Detection for Grounded Instruction Generation

♠**Lingjun Zhao** and ♣**Khanh Nguyen** and ♠◇**Hal Daumé III**
♠University of Maryland, College Park
♣University of California, Berkeley ◇Microsoft Research
lzhao123@umd.edu

## Abstract

We investigate the problem of generating instructions to guide humans to navigate in simulated residential environments. A major issue with current models is *hallucination*: they generate references to actions or objects that are inconsistent with what a human follower would perform or encounter along the described path. We develop a model that detects these hallucinated references by adopting a model pretrained on a large corpus of image-text pairs, and fine-tuning it with a contrastive loss that separates correct instructions from instructions containing synthesized hallucinations. Our final model outperforms several baselines, including using word probability estimated by the instruction-generation model, and supervised models based on LSTM and Transformer.

## 1 Introduction

Performance of neural-network-based models on generating navigation instructions is substantially inferior to that of humans (Zhao et al., 2023). These models often *hallucinate*, generating references to objects or actions that do not exist or are impossible to execute in the environment. Similar behavior has been observed in language models in other domains of text generation (Raunak et al., 2021; Ji et al., 2023; Xiao and Wang, 2021; Lee et al., 2018; Guerreiro et al., 2022; Rawte et al., 2023).

Instructions containing hallucinations can confuse or misdirect humans, leading to frustration and sometimes even catastrophic mistakes. Detecting hallucinations is therefore essential to improve instruction generation models and inform risk to human users. Nevertheless, ground-truth word-level hallucination labels are typically not readily available in this domain. Meanwhile, hiring crowdworkers to annotate instructions can be very costly (Anderson et al., 2018b; He et al., 2021; Wang et al., 2022; Gao et al., 2022).

We propose a data-efficient weakly supervised approach to hallucination detection. Our approach reduces the necessary supervision in two ways. First, we leverage a pre-trained vision-language model (Guhur et al., 2021) that has learned transferable representations of path-instruction pairs through self-supervised learning. Second, we introduce data-augmentation strategies to create synthetic data with "free" hallucination labels. We fine-tune the pre-trained model with the synthesized data using a contrastive learning objective to learn representations that separate positive examples (hallucinations) from negative examples (non-hallucinations). Our model outperforms various baselines in terms of F-1 scores on human-annotated evaluation data, beating an LSTM- and a Transformer-based models by 6.2 and 10.0 points, respectively. Ablation studies demonstrate the effectiveness of the proposed self-supervised pre-training and contrastive fine-tuning approach. We release the code, models, and data at https://lingjunzhao.github.io/hallucination_detection.html.

## 2 Related Work

**Hallucination detection.** Neural sequence to sequence models are prone to generate hallucinations, where the outputs are inconsistent with the inputs or the environments (Müller et al., 2019; Maynez et al., 2020; Wiseman et al., 2017; Martindale et al., 2019; Durmus et al., 2020; Ji et al., 2023). Recent work largely focuses on text-only domains (Wang and Sennrich, 2020; Zhou et al., 2020; Chen et al., 2021; Dale et al., 2022; Xu et al., 2023; Nie et al., 2019; Falke et al., 2019; Kryściński et al., 2019; Rebuffel et al., 2022; Liu et al., 2021; van der Poel et al., 2022) and image captioning (Rohrbach et al., 2018; Dai et al., 2022; Biten et al., 2022; Li et al., 2023; Gunjal et al., 2023). To the best of our knowledge, our work is the first study of hallucination in grounded instruction generation.

**Grounded Instruction Generation.** Instruction generation has been commonly studied in navigation settings (Anderson et al., 1991; Byron et al., 2010; Koller et al., 2010; Striegnitz et al., 2011; Goeddel and Olson, 2012; Fried et al., 2017, 2018; Wang et al., 2022; Kamath et al., 2022). Recent work by Zhao et al. (2023) reveals a significant gap between the performance of models and humans. Our work constructs a model that can be useful for evaluating and enhancing instruction-generation models. Huang et al. (2019) and Zhao et al. (2021) train LSTM-based discriminative models with contrastive learning to score instructions. We follow a similar approach but focus on identifying word-level hallucinations, and effectively leverage a large pre-trained Transformer model.

## 3 Problem Setting

**Grounded instruction generation.** Our task takes place in an environment, where a speaker model $S(\boldsymbol{u} \mid \boldsymbol{r})$ composes an *instruction* $\boldsymbol{u}$ to communicate an imaginary *trajectory* $\boldsymbol{r}$ to a follower so that the latter can generate the same trajectory in the environment. An instruction is a sequence of words $u_i$, whereas a trajectory is a sequence of observations $\boldsymbol{o}_t$ and actions $a_t$. We employ the Matterport3D simulator for experiments (Anderson et al., 2018b) which embeds a follower in a 3D model of a real-world residential building. The observation $\boldsymbol{o}_t$ of the follower comprises of an RGB image representing the panoramic view at a location in a building, and orientation features encoding the follower's gaze direction. Each action $a_t$ moves the follower to a new location close to where it is standing and changes its observation.

**Speaker model.** We follow Zhao et al. (2023) to train a T5-based (Raffel et al., 2020) speaker model. This model encodes a trajectory into a sequence of hidden vectors and applies multi-headed attention on those vectors to generate an instruction autoregressively. It is trained on the Room-to-Room (R2R) dataset provided by the Matterport3D simulator. Detail about the model is provided in §A.1.

**Hallucination in grounded instruction.** Instructions generated by our speaker model often contain words that are inconsistent with the input trajectory. We refer to those words as *hallucinations*. Similar to prior work (Zhou et al., 2020), we observe two types of hallucinations:

- *Intrinsic hallucination* is a word that needs to

be replaced because it inaccurately describes an observation or action. For example, an instruction says "*Walk past the reception desk and out the door on the right,*" but in the described trajectory, the door is on the *left*;

- *Extrinsic hallucination* is a word that needs to be removed because it has no correspondence in the input trajectory. Our model typically exhibits this type of hallucination by repeatedly generating the same sentence, e.g., "*Walk out of the office. Walk into the hallway and turn left. Walk into the hallway and turn left.*"

We formulate hallucination detection as *binary classification*: given an input $\boldsymbol{x} = (\boldsymbol{r}, \boldsymbol{u}, i)$ consisting of a trajectory $\boldsymbol{r}$, an instruction $\boldsymbol{u}$, and an index $i \in \{1, \cdots, |\boldsymbol{u}|\}$, decide whether the word $u_i$ is a hallucination, i.e. whether it should be replaced or removed to make $\boldsymbol{u}$ consistent with $\boldsymbol{r}$.

**Candidate selection.** For each instruction, we identify a set of candidate words for classification, which are (a) directional words like *left*, *right*, etc. (see §A.2 for a full list) as well as (b) nouns identified by the SpaCy part-of-speech tagger (Honnibal and Montani, 2017).

## 4 Hallucination Detection Model

### 4.1 Architecture

We learn a classifier $C(y = 1 \mid \boldsymbol{x} = (\boldsymbol{r}, \boldsymbol{u}, i))$ to decide whether a word $u_i$ is hallucinated. Our model is based on the Airbert model (Guhur et al., 2021), which inherits the ViLBERT architecture (Lu et al., 2019). An overview of the model is given in Figure 1. It implements two Transformers: one encodes the instruction $\boldsymbol{u}$ and the other encodes the trajectory $\boldsymbol{r}$. We wrap the word to be classified $u_i$ between a pair of special tokens ([BH] and [EH]). Let $\boldsymbol{h}_{\text{lang}}$ be the output of the language-encoding Transformer, and $\boldsymbol{h}_{\text{vision}}$ be that of the vision-encoding Transformer. The model computes a score function $s(\boldsymbol{x}) = s(\boldsymbol{r}, \boldsymbol{u}, i) = w^\top(\boldsymbol{h}_{\text{lang}} \odot \boldsymbol{h}_{\text{vision}})$, where $w$ is a learnable vector, and $\odot$ denotes element-wise multiplication. More details about the model are given in §A.1.

### 4.2 Learning approach

**Self-supervised pre-training.** Instead of learning from scratch, we fine-tune a pre-trained checkpoint of the Airbert model. The checkpoint was first trained on a large collection of 1.4M images

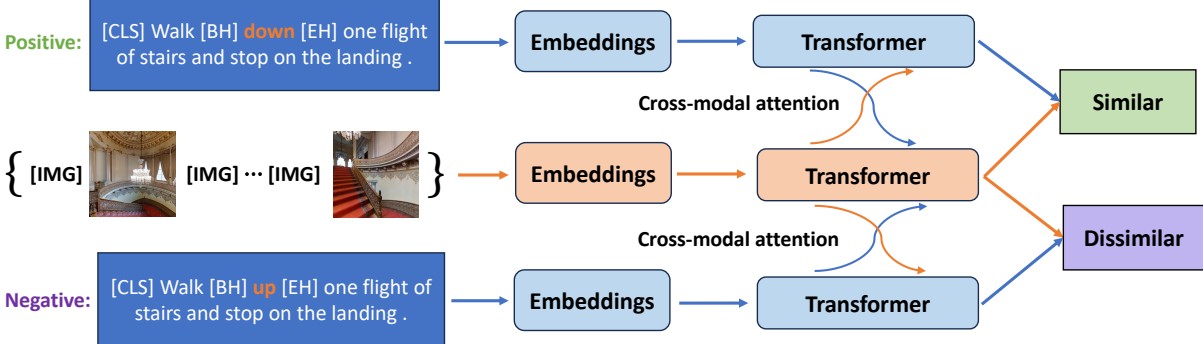

Figure 1: Our hallucination detection model, which takes as input an instruction with a target word and determines whether it should be replaced or removed to be consistent with a visual trajectory. To build this model, we fine-tune pre-trained Airbert (Guhur et al., 2021) with a contrastive learning objective.

and 0.7M captions collected from AirBnB. It was subsequently adapted for a trajectory-instruction compatibility estimation task using the Room-to-Room dataset. The objective in each phase combines BERT-style pre-training (mask and pair prediction) with contrastive learning. We refer the readers to the original paper for an elaborate description of the pre-training phase.

**Contrastive fine-tuning.** We assume a dataset of contrastive pairs $(\boldsymbol{x}^+, \boldsymbol{x}^-)$. The positive and negative examples of a pair have the same trajectory $\boldsymbol{r}$ and word index $i$, but differ in the instruction $\boldsymbol{u}$. The classified word in $\boldsymbol{x}^-$ is a hallucination, whereas that in $\boldsymbol{x}^+$ is not. For each pair, we compute the model scores $s(\boldsymbol{x}^+)$ and $s(\boldsymbol{x}^-)$, and construct the softmax distribution $\hat{\boldsymbol{p}} = \text{Softmax}(\boldsymbol{s})$ where $\boldsymbol{s} = (s(\boldsymbol{x}^+), s(\boldsymbol{x}^-))$. We then train the model to recognize the positive example by minimizing the cross entropy between $\hat{\boldsymbol{p}}$ and $\boldsymbol{p}^\star = (1, 0)$. This objective effectively forces the representation of the trajectory to be similar to that of the positive instruction and dissimilar to that of the negative instruction. At inference time, we define the hallucination detection classifier as $C(\boldsymbol{x}) = 1 - \sigma(s(\boldsymbol{x}))$, where $\sigma$ is the sigmoid function.

### 4.3 Synthesizing data creation

Even for fine-tuning, acquiring human-labeled data can be prohibitively expensive. For evaluation, we manually annotated a small sample of labels (§5). The annotation process was laborious, with an average time of 30 minutes required to annotate just 10 instructions. Based on our calculations, with a compensation of 15 USD per hour, it would cost approximately 9,000 USD to hire crowd workers to annotate all instances (∼12,000) in the R2R train-

ing set. Thus, we propose a more cost-effective methodology for generating training data.

**Synthetic negative examples.** We start with a training example $(\boldsymbol{u}^+, \boldsymbol{r})$ in the Room-to-Room training set and modify the human-written instruction $\boldsymbol{u}^+$ to create instructions with hallucinations. We first extract the candidate words in the instruction (§3). To create an intrinsic hallucination, we choose a candidate word and apply the following procedure:

- If the word is a **direction**, we replace it with an alternative direction. E.g., "*Walk ~~down~~ up one flight of stairs and stop on the landing.*";
- If it is a **room**, we substitute it with another room randomly selected from a pre-composed list. E.g., "*Exit the ~~bedroom~~ balcony via the farthest left. Walk toward the couch. Stop there.*";
- Otherwise, we swap it for another word in the instruction that is neither a direction nor a room. E.g., "*Exit the bedroom using the ~~door~~ step on the left then go straight until you get to the stairs and wait on the second ~~step~~ door.*"

Using this procedure, we first generate an intrinsic hallucination in $\boldsymbol{u}^+$ to synthesize $\boldsymbol{u}^-$. Then, with a probability of 0.5, we synthesize another intrinsic hallucination in each of $\boldsymbol{u}^+$ and $\boldsymbol{u}^-$. This step makes the training instructions more similar to the test-time inputs, which may contain multiple intrinsic hallucinations as they are generated by imperfect speaker models.

To create an instruction with *extrinsic* hallucinations, we append a sentence, taken from $\boldsymbol{u}^+$ or another instruction, to the end of a random sentence in $\boldsymbol{u}^+$. For example: "*Walk out of the office. Walk into the hallway and turn left. Walk*

| Model | F-1 | Precision | Recall |
|---|---|---|---|
| Random | 16.6 | 13.4 | 21.7 |
| Speaker model probability | 29.5 | 20.9 | 50.0 |
| LSTM-based encoder-decoder | 38.7 | 37.4 | 40.2 |
| T5-small (Transformer-based encoder-decoder) | 33.9 | 26.5 | 46.7 |
| T5-base (Transformer-based encoder-decoder) | 34.9 | 25.2 | **56.5** |
| Fine-tuned Airbert (ours) | **44.9** | **42.3** | 47.8 |

Table 1: Performance on the test set of our proposed hallucination detection model and various baselines. The decision threshold of each model is selected to maximize F-1 score of hallucination labels on the development set.

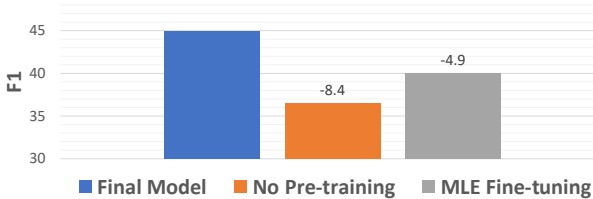

Figure 2: The effectiveness of self-supervised pre-training and contrastive fine-tuning. Results are F-1 scores of hallucination labels on the test set.

*into the hallway and turn left*.". Every word in the added sentence is considered an extrinsic hallucination. We do not create additional intrinsic hallucinations in the instruction.

**Alleviating input-distribution shift.** Model trained only on human-written instruction may perform poorly on model-generated instructions. Therefore, we also include "high-quality" model-generated instructions on the R2R training set as positive examples and apply the same strategies to generate negative examples. The quality of an instruction is measured by the success rate of an ensemble of VLN↺BERT instruction-following agents (Hong et al., 2021) in recreating the described trajectory. We consider a model-generated instruction to be of high quality if at least 80% of the ensemble agents can successfully reach the final location in the described trajectory.

## 5 Experiments

**Data.** Following the procedure described in §4.3, we generate a training set of 325,346 contrastive pairs. For evaluation, we use the same 75 evaluation trajectories in (Zhao et al., 2023) to form the test set. We randomly select another set of 20 trajectories in the R2R validation seen set for development. The environments in which the evaluation trajectories are generated are a subset of the training environments. We use the speaker model to generate instructions from these trajectories. The first two authors then manually annotate word-level hallucinations, creating 209 development examples and 632 test examples. The final labels are decided by mutual agreement. We choose the decision threshold of a model to maximize its F-1 score on the development set.

**Baselines.** (i) **random** classifier assigns a label chosen uniformly at random, (ii) **speaker model probability** defines the hallucination probability $C(\boldsymbol{x}) = 1 - S(u_i \mid \boldsymbol{r}; \boldsymbol{u}_{<i})$ where $\boldsymbol{x} = (\boldsymbol{r}, \boldsymbol{u}, i)$, $S$ is the speaker model (§ 3), and $\boldsymbol{u}_{<i}$ is the instruction generated up to step $i-1$ for the input $\boldsymbol{r}$; (iii) **LSTM** and (iv) **T5** are binary classifiers learned under a standard maximum-likelihood objective. They implement an encoder-decoder architecture based on LSTM and Transformer, respectively, and are trained using the same synthetic dataset as our proposed model. These models are initialized with random parameters. The detailed implementations and hyperparameters of all models are given in §A.1.

**Main results (Table 1).** The speaker-model-probability is a remarkably strong baseline, despite not trained for hallucination detection. Its performance is on par with that of T5, which is the same model but trained specifically for hallucination detection. The LSTM-based model outperforms the T5-based models. Scaling up the size of the T5 model improves the recall score by 10 points. Our proposed model (fine-tuned Airbert) beats all baselines by wide margins in terms of F-1 score for hallucination labels, (+10.0 versus T5-base, +6.2 versus LSTM). It excels in precision compared to the baselines. We also include results on the development set in §A.3.

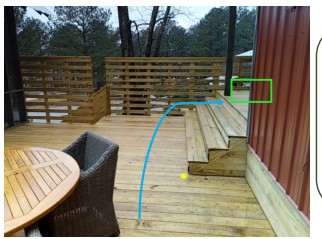

(a) Success on detecting extrinsic hallucination: the second sentence should be removed entirely; the model marks all the candidate words in the sentence.

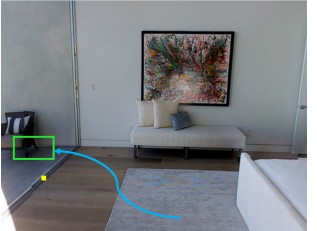

(b) Success on detecting intrinsic hallucination: the correct direction is to go to the left side of the bedroom, not exiting it.

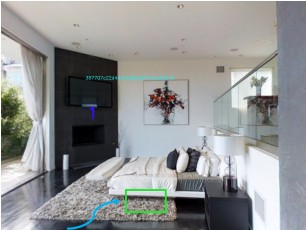

(c) Model misidentifies the stopping location due to lacking depth information: the TV in the far left corner looks to be close to the true stopping location.

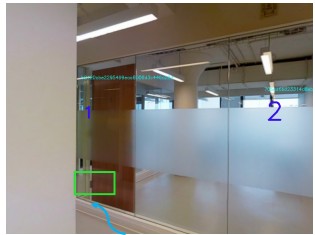

(d) Ambiguous direction: a slight left turn that appears like a straight walk in this viewpoint.

Figure 3: Some successful and failure cases of the fine-tuned Airbert model. The blue arrow indicates the described path, and the green represents the next location.

**Ablation studies (Figure 2).** Our results confirm that self-supervised pre-training and contrastive fine-tuning are requisite to the performance of our model. Without pre-training, our model is just as bad as the LSTM-based model. We also compare fine-tuning via contrastive learning with fine-tuning via a maximum-likelihood learning. In the latter approach, the model simply takes as input an example $(r, u, i)$ and learns to directly predict the true label. The approach underperforms contrastive learning by 4.9 F-1 points. Our finding aligns with previous work (Gunel et al., 2021; Zhang et al., 2021; Goyal et al., 2023), suggesting that contrastive learning is effective not only as a representation learning objective, but also as a classification objective.

**Error and Qualitative Analysis.** In Table 2, we break down the performance of our model by word type. Our model struggles with detecting room and object hallucinations, indicating that its understanding of visually grounded words is lacking. Especially, it has relatively low recall on object hallucinations, potentially due to lack of diversity of this word type in the training data. Figure 3 shows a few successful and failure examples of our model.

## 6 Conclusion

This work is an early attempt to address the hallucination issue in grounded instruction generation. We have shown that techniques like self-supervised

| Type | F1 | Precision | Recall |
|------|------|------|------|
| Direction | 48.1 | 41.9 | 56.4 |
| Room | 38.9 | 38.9 | 38.9 |
| Object | 38.7 | 50.0 | 31.6 |

Table 2: Fine-tuned Airbert performance broken down by word type. Results are on test set.

pre-training on multimodal data and contrastive fine-tuning on synthetic data are promising scalable approaches. We hope that these directions can be further developed in future work.

## Limitations

Despite the effectiveness of the data generation method, this approach requires substantial domain-specific knowledge. Our method, particularly to generate directional hallucinations, is based on heuristics and does not take into account the actual environment. Another limitation is the small size of the evaluation datasets due to the expensive cost of annotation.

## Acknowledgments

We thank the CLIP Laboratory at Maryland and our reviewers for providing helpful feedback to improve the manuscript.

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

## A  Appendices

### A.1  Models

**Speaker.** The speaker model takes as input a trajectory and computes a distribution over instructions. To encode a trajectory $r$, following prior work (Shen et al., 2021; Zhao et al., 2023), we convert each panoramic observation $o_t$ into a collection of 36 images that represent the first-person views obtained from 36 gaze directions. We feed these into a pre-trained vision model (Radford et al., 2021) to obtain a set of view vectors $\{o_t^i\}_{i=1}^{36}$. For each action $a_t$, which corresponds to an adjacent location, let $k \in \{1, \cdots, 36\}$ be the direction towards that location and let $\theta = (\theta^{\text{hor}}, \theta^{\text{ver}})$ be the horizontal and vertical angles of that direction. We represent $a_t$ by concatenating the visual features $o_t^k$ with the directional features $[\cos\theta, \sin\theta]$. The sequence of observation and action representations is fed into a Transformer encoder to produce a sequence of hidden vectors. A transformer decoder then applies multi-headed attention to those vectors and generates an instruction $u$ auto-regressively.

**Airbert model.** The input of the classifier is a trajectory $r$ and an instruction $u$. The instruction has the following format:

$$\left[\, [\text{CLS}], u_1, \ldots, [\text{BH}], u_i, [\text{EH}], \ldots, u_{|\boldsymbol{u}|}, [\text{SEP}] \,\right]$$

where the word to be classified $u_i$ is enclosed by special tokens [BH] and [EH], and the [CLS] and [SEP] tokens mark the beginning and the end. Each token are replaced by a sum of a token embedding and a positional embedding. We pass this sequence of embeddings into a Transformer.

For the trajectory, the model extracts from a panoramic view $o_t$ a set of image regions $\{o_t^{(j)}\}_{j=1}^{K}$ and represents the sequence of observations as:

$$\left[\, [\text{IMG}], o_1^{(1)}, \ldots, o_1^{(K)}, [\text{IMG}], o_2^{(1)}, \ldots, o_2^{(K)}, \right.$$
$$\left. \cdots, [\text{IMG}], o_T^{(1)}, \ldots, o_T^{(K)} \,\right]$$

where [IMG] is the embedding of an observation-separating token. Each image region $o_t^{(j)}$ is converted into a visual embedding, which is an addition of three embeddings: visual embedding (computed by a Faster R-CNN model (Anderson et al., 2018a)), directional embedding, and region-index embedding. We feed the sequence of visual embeddings into a second Transformer.

Let $h_{\text{lang}}$ be the output at the position of the [CLS] token of the language-encoding Transformer, and $h_{\text{vision}}$ be the output at the position of the first [IMG] token of the vision-encoding Transformer. The score function $s(x)$ is defined as:

$$s(\boldsymbol{x}) = s(\boldsymbol{r}, \boldsymbol{u}, i) = w^\top (h_{\text{lang}} \odot h_{\text{vision}})$$

where $w$ is a learnable vector, and $\odot$ denotes element-wise multiplication.

**T5.** This model is the same as the speaker model. However, instead of generating an instruction, it computes a score $s(x)$ like the Airbert model. The input $x$ is also a tuple $(r, u, i)$. The instruction $u$ has the same format as in the case of the Airbert model, with the word to be classified surrounded by two special tokens. Let $\{h_j\}_{j=1}^{|\boldsymbol{u}|}$ be the sequence of hidden vectors obtain after decoding the input instruction. We compute the mean vector $h = \frac{1}{|\boldsymbol{u}|} \sum_{j=1}^{|\boldsymbol{u}|} h_j$. The score is computed as $s(\boldsymbol{x}) = w^\top h$, where $w$ is a learnable vector.

**LSTM.** This model is similar to the T5 model except that the encoder and decoder are LSTMs.

**Hyperparamters and Computation.** The hyperparameters and computation cost of all models are listed in Table 3.

### A.2  Word replacement

We compiled a list of direction words and divided them into groups (Table 4). When constructing a negative example, if a word is selected, a replacement is randomly selected among the remaining words in the same group.

Our compiled list of rooms to generate synthetic examples are: { "laundry room", "mudroom", "family room", "balcony", "utility room", "tool room", "entryway", "foyer", "lobby", "library", "bathroom", "bar", "spa", "sauna", "living room", "other room", "staircase", "garage", "hallway", "office", "classroom", "outdoor areas", "meeting room", "conference room", "dining room", "lounge", "bedroom", "porch", "terrace", "deck", "driveway", "kitchen", "toilet", "workout room", "exercise room", "gym", "tv room", "recreation room ", "game room", "closet", "junk", "study", "guest room", "music room", "home theater", "sunroom", "conservatory", "playroom", "pantry",

| Hyperparameters | Fine-tuned Airbert | T5 | LSTM | Speaker Model |
|---|---|---|---|---|
| Learning rate | $10^{-5}$ | $10^{-4}$ | $10^{-4}$ | $10^{-4}$ |
| Batch size | 128 | 64 | 64 | 32 |
| Optimizer | AdamW | AdamW | AdamW | AdamW |
| Num. of training iterations | $5 \times 10^5$ | $6 \times 10^5$ | $3 \times 10^5$ | $16 \times 10^4$ |
| Max. instruction length | 60 | 80 | 80 | 80 |
| Image feature size | 2048 | 512 | 512 | 512 |
| Embedding dropout | 0.1 | 0.3 | 0.3 | 0.3 |
| Hidden size | 768 | 512 | 512 | 512 |
| Num. of Transformer/LSTM layers | 12 | 4 | 2 | 4 |
| Transformer/LSTM dropout rate | 0.1 | 0.2 | 0.5 | 0.3 |
| Num. of parameters (million) | 250M | 57M (small), 120M (base) | 8M | 57M |
| Computation and training time | RTX A4000: 72h | RTX A6000: 72h | RTX A6000: 48h | RTX A6000: 48h |
| Hallucination Threshold | 0.92 | 0.98 | 0.98 | 0.42 |

Table 3: The hyperparameters of all models. For the T5 models, we use decoders with two layers, which improve the performance compared to the original decoders.

| Direction Type | Candidate Words | | |
|---|---|---|---|
| **Horizontal** | left | right | |
| | front | back | |
| | forward | backward | |
| | towards | away from | |
| | through | past | |
| | leftmost | rightmost | |
| **Vertical** | bottom | middle | top |
| | up | down | |
| | above | under | |
| **Location** | enter | exit | |
| | into | out of | |
| | inside | outside | |
| | first | second | third |

Table 4: Directional word list. Each row shows a group of words.

"storage room", "attic", "basement", "gallery", "greenhouse", "yoga studio", "meditation room", "stairs", "staircase", "floor" }. These are based on room labels in the Matterport3D dataset (Chang et al., 2017) and suggestions of GPT-4 (OpenAI, 2023).

## A.3 Results on development set (Table 5)

| Model | F-1 | Precision | Recall |
|---|---|---|---|
| Random | 20.4 | 20.0 | 20.8 |
| Speaker probability | 45.5 | 37.3 | 58.3 |
| LSTM | 34.0 | 32.7 | 35.4 |
| T5-small | 44.9 | 34.4 | 64.6 |
| T5-base | 40.0 | 28.6 | **66.7** |
| Fine-tuned Airbert | **57.1** | **50.0** | **66.7** |

Table 5: Performance on the development set of all models.