# OpenReview forum: "Hallucination Detection for Grounded Instruction Generation"
_EMNLP/2023/Conference — EMNLP 2023 Findings_

### Official Review · Reviewer_Kkpf · 2023-08-01

**Soundness:** 3

**Excitement:**

3: Ambivalent: It has merits (e.g., it reports state-of-the-art results, the idea is nice), but there are key weaknesses (e.g., it describes incremental work), and it can significantly benefit from another round of revision. However, I won't object to accepting it if my co-reviewers champion it.

**Paper Topic And Main Contributions:**

Instruction generation models in embodied navigation tasks tend to hallucinate by generating inconsistent actions or repeating themselves. This paper proposes to detect such hallucinations using a model trained in a joint pre-training + contrastive-finetuning fashion. To improve contrastive learning, the author(s) propose using synthetic negatives generated by altering the rooms/directions or swapping words in generated instructions. On the R2R dataset, the author(s) demonstrate their objectives applied to a pre-trained Airbert model detects hallucinations better than models trained using the standard likelihood objective.

**Questions For The Authors:**

- Could the authors clarify if pre-training is a novelty of this paper? If so, in what way is it novel?
- What does the performance for the baselines look like once they are parameter-matched? (say, by using a T5-base instead of T5-small)

**Reasons To Accept:**

The approach of constructing synthetic negatives by swapping rooms/directions is interesting, particularly based on the amount of improvement observed from the addition of such negatives.

**Reasons To Reject:**

- Pre-training a transferred model on the new task is claimed to be a novelty here. However, task-adaptive pre-training has been long known to help improve any classification task (Gururangan et al. 2020). This limits the novelty of the proposed pre-training strategy in this work. From the result in Table 2, without pre-training, the fine-tuned Airbert model performs almost as well as the LSTM model (worse than the Speaker and the T5 models)
- The baselines in Table 1 are significantly weaker than the proposed model in terms of model parameters (57 million for speaker/T5 vs 250 million for Airbert from Appendix A.1; lines 649-657) which may skew the results in favor of the model shown here. Without a clear comparison with baselines that are nearly parameter-matched, these numbers do not add much value to the analysis.

**Reference**:

Gururangan et al. 2020: Don't Stop Pretraining: Adapt Language Models to Domains and Tasks, ACL 2020

**Reproducibility:**

3: Could reproduce the results with some difficulty. The settings of parameters are underspecified or subjectively determined; the training/evaluation data are not widely available.

**Reviewer Confidence:**

4: Quite sure. I tried to check the important points carefully. It's unlikely, though conceivable, that I missed something that should affect my ratings.

---

> ### Author Rebuttal · Authors · 2023-08-28
>
> We thank you for your thorough review and questions.
>
> **“Is pre-training a novelty of this paper?”**: Pre-training method is not novel, but applying it to this problem is novel. While pre-training has been shown to help on a variety of classification tasks, any claim that it would help on a new task needs to be empirically verified. This task is especially challenging than standard text or image classification tasks: it involves understanding the correspondence between a **sequence of photos** and a **sequence of words**. The novelty of our work also lies in contrastively fine-tuning a pre-trained model using a synthetically created dataset.
>
> **“Comparison with parameter-matched baselines”**: We trained a hallucination detection model using T5-base, to have another row of experimental results for Table 1. On the Test set, T5-base hallucination detection model underperforms our model by 32.4% while having a comparable number of parameters (F1=34.9). On the Dev set, T5-base model underperforms our model by 42.8% (F1=40.0). As the training data is not very large, using a larger model might not be too helpful here. The T5-base hallucination detection model still underperforms our proposed model.

---

### Official Review · Reviewer_CKdk · 2023-08-04

**Soundness:** 3

**Excitement:**

2: Mediocre: This paper makes marginal contributions (vs non-contemporaneous work), so I would rather not see it in the conference.

**Paper Topic And Main Contributions:**

The paper presents a model for detecting hallucinated words in navigation instructions. They present a formulation of the hallucination detection problem at the word-level, create new evaluation data, and establish baselines.

**Questions For The Authors:**

- A. Why would repeating a sentence lead to external hallucination?
- B. Line 201: "we manually annotated a small sample of labels." How many? Was this data used in the synthetic data creation, or was it done just to see how time-consuming it would be?
- C. §5: Are the LSTM and T5 baselines trained on the same synthetic dataset?
- D. §5: Based on my understanding, since the task is to evaluate whether a given word in an instruction is hallucinated, a different index for the same instruction would count as a distinct example. If there are 632 test examples, how many instructions do they come from? Did you use every word in each instruction as an example, or only words susceptible to hallucination with the same method in §3? How many authors helped to annotate this data, and what was inter-annotator agreement?
- E. §5: Why was the dev set collected? It's not clear how it was used.
- F. Do you plan to release the dev and test data that you created?

**Reasons To Accept:**

The paper tackles an interesting problem. Hallucination is one of the most prominent issues in NLP today, and it is interesting to see discussion of it in the multimodal space.

**Reasons To Reject:**

- The method is evaluated on one test set, created by the authors manually annotating a small subset (75 trajectories) of an existing dataset (R2R). They produce 632 test examples this way, but it is not clear how many unique instructions this corresponds to. As seen in Table 2, results vary greatly between the dev and test set, and there is no reporting of variance across different random seeds for training.
- The gold labels in the test set are annotated by the authors themselves. There is no description of how many authors did annotation, or what inter-annotator agreement was.
- The evaluation setting is limited in terms of impact. The paper would be much stronger if the authors showed that if they use their method to discard hallucinated instructions, then they can improve performance on an existing benchmark.
- The synthetic data creation method is relatively limited, with three word-swap rules for creating intrinsic hallucinations, and appending a random sentence for creating extrinsic hallucinations.

**Reproducibility:**

3: Could reproduce the results with some difficulty. The settings of parameters are underspecified or subjectively determined; the training/evaluation data are not widely available.

**Reviewer Confidence:**

3: Pretty sure, but there's a chance I missed something. Although I have a good feel for this area in general, I did not carefully check the paper's details, e.g., the math, experimental design, or novelty.

**Typos Grammar Style And Presentation Improvements:**

- I wonder if the bigram "navigation instructions" should be in the title, to distinguish from instructions for models, as in the instruction-tuning literature.
- Lines 147-148: "For each instruction, we pre-select a subset of words as candidates for classification." This makes it sound like *for each instruction*, you manually identified candidate words. I would suggest "We develop heuristics to identify candidate words, which we apply to all instructions" (if it is true).

---

> ### Author Rebuttal · Authors · 2023-08-28
>
> We thank you for your thorough assessment and suggestions. To answer the questions,
>
> **Question A**: For example, if we add a repeating sentence to ground truth instruction A “Walk out of the office. Walk into the hallway and turn left”, the new instruction B becomes “Walk out of the office. Walk into the hallway and turn left. **Walk into the hallway and turn left**”. The last sentence either does not make sense in the environment after completing the second sentence, or the last sentence would lead to a wrong destination. Each word in the last sentence from instruction B needs to be removed in order to be consistent with the input trajectory, thus repeating a sentence most likely will lead to external hallucination during the synthetic dataset creation.
>
> **Question B**: We annotated 209 development examples and 632 test examples (mentioned in Section 5 line 255), in order to evaluate our proposed model and baseline models. The datasets were created for evaluation, instead of training.
>
> **Question C**: Yes, the baselines were trained on the same synthetic dataset as our proposed model.
>
> **Question D**: The 632 test examples come from 75 instructions. We use only words that are susceptible to hallucination with the same method in §3. Two authors annotated the datasets, and the labels are determined by mutual agreement.
>
> **Question E**: The dev set was collected for determining the hallucination decision threshold of each model by maximizing its F-1 score (Table 1 caption). If a word has a hallucination probability greater than the selected threshold, the word is considered hallucination.
>
> **Question F**: Yes, we plan to release the datasets.
>
> **“Results vary greatly between the dev and test set in Table 2”**: The dev set is small, and this dataset was created for determining the hallucination decision threshold of each model by maximizing its F-1 score. We consider the results on the test set to be more reliable as this dataset is larger.
>
> **“The evaluation setting is limited in terms of impact”**: Other evaluation settings to compare with existing benchmarks, e.g.discarding hallucinated instructions to improve performance, can be an interesting follow-up. Human evaluation by navigation with the generated instruction is necessary in those settings. We experimented with the evaluation setting of communicating hallucination to human users, however, it is a non-trivial problem. Our preliminary results show that humans are susceptible to the text instructions and the display of the instruction. We will investigate these problems in future work.
>
> For your other suggestions, we will incorporate them in the next version.

---

### Official Review · Reviewer_ieLz · 2023-08-12

**Soundness:** 4

**Excitement:**

3: Ambivalent: It has merits (e.g., it reports state-of-the-art results, the idea is nice), but there are key weaknesses (e.g., it describes incremental work), and it can significantly benefit from another round of revision. However, I won't object to accepting it if my co-reviewers champion it.

**Paper Topic And Main Contributions:**

### A summary of the paper
- The paper investigates detecting hallucinations (inconsistent references to objects or actions) in instructions generated by models to navigate simulated environments (Section 1).

- The authors categorize the hallucinations into *intrinsic hallucinations* -- inaccurate description of the trajectory that needs to be replaced by a correct word and *extrinsic hallucinations* -- extra unnecessary words that should be removed as they are unrelated to the trajectory. (Section 3)

- The paper proposes a weakly-supervised approach using a pre-trained vision-language model fine-tuned with synthetic data to detect hallucinations (Sections 3-4).

- The approach uses data augmentation strategies to create synthetic training data with free hallucination labels (Section 4.3).

- A contrastive learning objective is used to fine-tune the pre-trained model to distinguish hallucinations from non-hallucinations (Section 4.2).

- The proposed model outperforms LSTM and Transformer baselines and ablation studies show benefits of pre-training and contrastive fine-tuning (Section 5).

- The model struggles with some types of hallucinations like detecting rooms and objects, due to lack of diversity in training data (Section 5).

### Contributions
The core contributions are developing a weakly-supervised and data-efficient approach for hallucination detection for grounded instruction following, and showing its effectiveness compared to supervised baselines. The pre-training plus contrastive fine-tuning method helps reduce the need for expensive human annotations.

**Reasons To Accept:**

1. **Importance of the problem**: A system that is describe data in text form often exhibits *hallucination* problem. Despite that this is a huge problem in the general domain, a focus study on *Grounded Instruction Following* is still useful to Embodied AI and pragmatics community.
2. **The innovation of the approach**: Creating synthetic data and training with contrastive objective are not new techniques, but afaik, they are used in this problem for the first time.
3. **Experiments**: The results show that their model and training method is better than the baselines.

**Reasons To Reject:**

My major concern is that **Only two kinds of hallucination are studied**: using wrong word or repetitive sentences according to their data generation method. In practice the hallucination could be more complicated, e.g. instruction for going a detour in the middle (which could be part of the trajectory [1]) or asking the model to do a subtask that doesn't exist in the trajectory. More insights should be given regarding approaches and empirical results regarding finding this kind of hallucination.

------
EDIT after rebuttal: I have read the author's rebuttal and other reviews. Reviewer CKdk raised a similar concern regarding synthetic data creation method in this paper. I think this paper will be much stronger if the authors include other kinds of hallucination in both training and test data. The author's rebuttal was insufficient to resolve this concern.

I will lower my excitement score to 3 (*it can significantly benefit from another round of revision*), but keep the soundness score (*This study provides sufficient support for all of its claims/arguments*, although the claims are weak).

------
[1]: Don't Copy the Teacher: Data and Model Challenges in Embodied Dialogue. EMNLP 2022. So Yeon Min, Hao Zhu, Ruslan Salakhutdinov, Yonatan Bisk

**Reproducibility:**

4: Could mostly reproduce the results, but there may be some variation because of sample variance or minor variations in their interpretation of the protocol or method.

**Reviewer Confidence:**

4: Quite sure. I tried to check the important points carefully. It's unlikely, though conceivable, that I missed something that should affect my ratings.

---

> ### Author Rebuttal · Authors · 2023-08-28
>
> We thank you for your thorough review and suggestions.
>
> **Only two kinds of hallucination are studied**: We create synthetic extrinsic hallucinations by not only repeating sentences, but also randomly sample a sentence from the instruction dataset to add to the ground truth instruction, which is to ask the model to do a subtask that most likely does not exist in the trajectory. We agree that it would be interesting to investigate going a detour in the middle, and will incorporate the suggestions in future work.

---

### Meta-Review · Area_Chair_AXwL · 2023-09-18

**Recommendation:** 3

**Metareview:**

Reviewers agreed that the paper was sufficiently sound, but generally agreed that the scope of the paper was limited. Reviewers raised concerns about the evaluation task (only hallucination detection, rather than downstream performance on VLN after filtering augmented data, or instruction generation), small size of the test dataset, and limited type of hallucinations detected. However, reviewers also identified that the problem is important, and the method is simple (uses a small number of word-swapping rules, and random sentence appending, in combination with pre-training) but effective. This method, or extensions of it, may prove helpful in some downstream tasks.

One additional paper on hallucination detection for generation in VLN settings is Huang et al., Multi-Modal Discriminative Model for Vision-and-Language Navigation. SpLU-RoboNLP workshop, 2019, although it doesn't diminish the novelty of this paper as it keeps the language the same and corrupts paths, and doesn't use pre-training.

---

### Decision · Program_Chairs · 2023-10-07

**Decision:**

Accept-Findings

**Comment:**

Reviewers agreed that the paper was sufficiently sound, but generally agreed that the scope of the paper was limited. Reviewers raised concerns about the evaluation task (only hallucination detection, rather than downstream performance on VLN after filtering augmented data, or instruction generation), small size of the test dataset, and limited type of hallucinations detected. However, reviewers also identified that the problem is important, and the method is simple (uses a small number of word-swapping rules, and random sentence appending, in combination with pre-training) but effective. This method, or extensions of it, may prove helpful in some downstream tasks.

One additional paper on hallucination detection for generation in VLN settings is Huang et al., Multi-Modal Discriminative Model for Vision-and-Language Navigation. SpLU-RoboNLP workshop, 2019, although it doesn't diminish the novelty of this paper as it keeps the language the same and corrupts paths, and doesn't use pre-training.